# Faithful Inversion of Generative Models for Effective Amortized Inference

**Stefan Webb**[*]
University of Oxford

**Adam Goliński**
University of Oxford

**Robert Zinkov**
UBC

**N. Siddharth**
University of Oxford

**Tom Rainforth**
University of Oxford

**Yee Whye Teh**
University of Oxford

**Frank Wood**
UBC

## Abstract

Inference amortization methods share information across multiple posterior-inference problems, allowing each to be carried out more efficiently. Generally, they require the inversion of the dependency structure in the generative model, as the modeller must learn a mapping from observations to distributions approximating the posterior. Previous approaches have involved inverting the dependency structure in a heuristic way that fails to capture these dependencies correctly, thereby limiting the achievable accuracy of the resulting approximations. We introduce an algorithm for faithfully, and minimally, inverting the graphical model structure of any generative model. Such inverses have two crucial properties: a) they do not encode any independence assertions that are absent from the model and b) they are local maxima for the number of true independencies encoded. We prove the correctness of our approach and empirically show that the resulting minimally faithful inverses lead to better inference amortization than existing heuristic approaches.

## 1 Introduction

Evidence from human cognition suggests that the brain reuses the results of past inferences to speed up subsequent related queries (Gershman & Goodman, 2014). In the context of Bayesian statistics, it is reasonable to expect that, given a generative model, $p(\mathbf{x}, \mathbf{z})$, over data $\mathbf{x}$ and latent variables $\mathbf{z}$, inference on $p(\mathbf{z} \mid \mathbf{x}_1)$ is informative about inference on $p(\mathbf{z} \mid \mathbf{x}_2)$ for two related inputs, $\mathbf{x}_1$ and $\mathbf{x}_2$. Several algorithms (Kingma & Welling, 2014; Rezende et al., 2014; Stuhlmüller et al., 2013; Paige & Wood, 2016; Le et al., 2017, 2018; Maddison et al., 2017a; Naesseth et al., 2018) have been developed with this insight to perform *amortized inference* by learning an inference artefact $q(\mathbf{z} \mid \mathbf{x})$, which takes as input the values of the observed variables, and—typically with the use of neural network architectures—return a distribution over the latent variables approximating the posterior. These inference artefacts are known variously as inference networks, recognition models, probabilistic encoders, and guide programs; we will adopt the term *inference networks* throughout.

Along with conventional fixed-model settings (Stuhlmüller et al., 2013; Le et al., 2017; Ritchie et al., 2016; Paige & Wood, 2016), a common application of inference amortization is in the training of variational auto-encoders (VAEs) (Kingma & Welling, 2014), for which the inference network is simultaneously learned alongside a generative model. It is well documented that deficiencies in the expressiveness or training of the inference network can also have a knock-on effect on the learned generative model in such contexts (Burda et al., 2016; Cremer et al., 2017, 2018; Rainforth et al., 2018), meaning that poorly chosen coarse-grained structures can be particularly damaging.

Implicit in the factorization of the generative model and inference network in both fixed and learned model settings are probabilistic graphical models, commonly Bayesian networks (BNs), encoding dependency structures. We refer to these as the *coarse-grain* structure, in opposition to the *fine-grain* structure of the neural networks that form each inference (and generative) network factor. In this sense, amortized inference can be framed as the problem of graphical model inversion—how to invert the graphical model of the generative model to give a graphical model approximating the posterior.

---

[*]Correspondence to `info@stefanwebb.me`

Many models from the deep generative modeling literature can be represented as BNs (Krishnan et al., 2017; Gan et al., 2015; Neal, 1990; Kingma & Welling, 2014; Germain et al., 2015; van den Oord et al., 2016b,a), and fall within this framework.

In this paper, we borrow ideas from the probabilistic graphical models literature, to address the previously open problem of how best to automate the design of the coarse-grain structure of the inference network (Ritchie et al., 2016). Typically, the inverse graphical model is formed heuristically. At the simplest level, some methods just invert the edges in the BN for the generative model, removing edges between observed variables (Kingma & Welling, 2014; Gan et al., 2015; Ranganath et al., 2015). In a more principled, but still heuristic, approach, Stuhlmüller et al. (2013); Paige & Wood (2016) construct the inference network by inverting the edges and additionally connecting the parents of children in the original graph (both of which are a subset of a variable's Markov blanket; see Appendix C).

In general, these heuristic methods introduce conditional independencies into the inference network that are not present in the original distribution. Consequently, they cannot represent the true posterior even in the limit of infinite neural network capacities. Take the simple generative model with branching structure of Figure 1a. The inference network formed by Stuhlmüller's method inverts the edges of the model as in Figure 1b. However, an inference network that is able to represent the true posterior requires extra edges between the branches, as in Figure 1c.

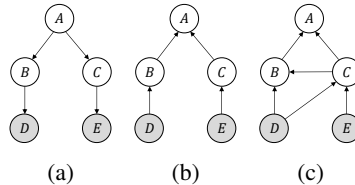

(a)      (b)      (c)

Figure 1: (a) Generative model BN; (b) Inverse BN by Stuhlmüller's Algorithm; (c) *Faithful* inverse BN by our algorithm.

Another approach, taken by Le et al. (2017), is to use a fully connected BN for the inverse graphical model, such that every random choice made by the inference network depends on every previous one. Though such a model is expressive enough to correctly represent the data given infinite capacity and training time, it ignores substantial available information from the forward model, inevitably leading to reduced performance for finite training budgets and/or network capacities.

In this paper, we develop a tractable framework to remedy these deficiencies: the *Na*tural *M*inimal *I*-map generator (NaMI). Given an arbitrary BN structure, NaMI can be used to construct an inverse BN structure that is provably both *faithful* and *minimal*. It is faithful in that it contains sufficient edges to avoid encoding conditional independencies absent from the model. It is minimal in that it does not contain any unnecessary edges; i.e., removing any edge would result in an unfaithful structure.

NaMI chiefly draws upon variable elimination (Koller & Friedman, 2009, Ch 9,10), a well-known algorithm from the graphical model literature for performing exact inference on discrete factor graphs. The key idea in the operation of NaMI is to simulate variable elimination steps as a tool for successively determining a minimal, faithful, and natural inverse structure, which can then be used to parametrize an inference network. NaMI further draws on ideas such as the min-fill heuristic (Fishelson & Geiger, 2004), to choose the ordering in which variable elimination is simulated, which in turn influences the structure of the generated inverse.

To summarize, our key contributions are:

  i) framing generative model learning through amortized variational inference as a graphical model inversion problem, and

 ii) using the simulation of exact inference algorithms to construct an algorithm for generating provably minimally faithful inverses.

Our work thus highlights the importance of constructing both minimal and faithful inverses, while providing the first approach to produce inverses satisfying these properties.

## 2 Method

Our algorithm builds upon the tools of *probabilistic graphical models*— a summary for unfamiliar readers is given in Appendix A.

### 2.1 General idea

Amortized inference algorithms make use of inference networks that approximate the posterior. To be able to represent the posterior accurately, the distribution of the inference network should not encode independence assertions that are absent from the generative model. An inference network that did

encode additional independencies could not represent the true posterior, even in the non-parametric limit, with neural network factors whose capacity approaches infinity.

Let us define a *stochastic inverse* for a generative model $p(\mathbf{x}|\mathbf{z})p(\mathbf{z})$ that factors according to a BN structure $\mathcal{G}$ to be a factorization of $q(\mathbf{z}|\mathbf{x})q(\mathbf{x})$ over $\mathcal{H}$ (Stuhlmüller et al., 2013; Paige & Wood, 2016). The $q(\mathbf{z}|\mathbf{x})$ part of the stochastic inverse will define the factorization, or rather, coarse-grain structure, of the inference network. Recall from §1 that this involved two characteristics. We first require $\mathcal{H}$ to be an *I-map* for $\mathcal{G}$:

**Definition 1.** *Let $\mathcal{G}$ and $\mathcal{H}$ be two BN structures. Denote the set of all conditional independence assertions made by a graph, $\mathcal{K}$, as $\mathcal{I}(\mathcal{K})$. We say $\mathcal{H}$ is an* I-map *for $\mathcal{G}$ if $\mathcal{I}(\mathcal{H}) \subseteq \mathcal{I}(\mathcal{G})$.*

To be an I-map for $\mathcal{G}$, $\mathcal{H}$ may not encode all the independencies that $\mathcal{G}$ does, but it must not mislead us by encoding independencies not present in $\mathcal{G}$. We term such inverses as being *faithful*. While the aforementioned heuristic methods *do not* in general produce faithful inverses, using either a fully-connected inverse, or our method, does.

Second, since a fully-connected graph encodes no conditional independencies and is therefore suboptimal, we require in addition that $\mathcal{H}$ be a *minimal I-map* for $\mathcal{G}$:

**Definition 2.** *A graph $\mathcal{K}$ is a* minimal I-map *for a set of independencies $\mathcal{I}$ if it is an I-map for $\mathcal{I}$ and if removal of even a single edge from $\mathcal{K}$ renders it not an I-map.*

We call such inverses *minimally faithful*, which roughly means that the inverse is a local optimum in the number of true independence assertions it encodes.

There will be many minimally faithful inverses for $\mathcal{G}$, each with a varying number of edges. Our algorithm produces a *natural inverse* in the sense that it either inverts the order of the random choices from that of the generative model (when it is run in the topological mode), or it preserves the ordering of the random choices (when it is run in reverse topological mode):

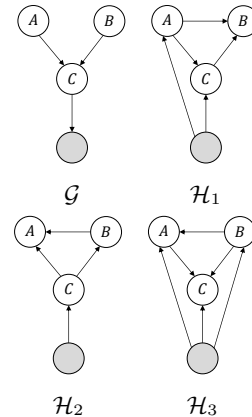

**Definition 3.** *A stochastic inverse $\mathcal{H}$ for $\mathcal{G}$ over variables $\mathcal{X}$ is a* natural inverse *if either, for all $X \in \mathcal{X}$ there are no edges in $\mathcal{H}$ from $X$ to its descendants in $\mathcal{G}$, or, for all $X \in \mathcal{X}$ there are no edges in $\mathcal{H}$ from $X$ to its ancestors in $\mathcal{G}$.*

Essentially, a natural inverse is one for which if we were to perform ancestral sampling, the variables would be sampled in either a topological or reverse-topological ordering, relative to the original model. Consider the inverse networks of $\mathcal{G}$ shown in Figure 2. $\mathcal{H}_1$ is not a natural inverse of $\mathcal{G}$, since there is both an edge $A \to C$ from a parent to a child, and an edge $C \to B$ from a child to a parent, relative to $\mathcal{G}$. However, $\mathcal{H}_2$ and $\mathcal{H}_3$ are natural, as they correspond respectively to the reverse-topological and topological orderings $C, B, A$ and $B, A, C$.

Most heuristic methods, including those of (Stuhlmüller et al., 2013; Paige & Wood, 2016), produce (unfaithful) natural inverses that invert the order of the random choices, giving a reverse-topological ordering.

Figure 2: Illustrating definition of naturalness.

## 2.2   Obtaining a natural minimally faithful inverse

We now present NaMI's graph inversion procedure that given an arbitrary BN structure, $\mathcal{G}$, produces a natural minimal I-map, $\mathcal{H}$. We illustrate the procedure step-by-step on the example given in Figure 3. Here $H$ and $J$ are observed, as indicated by the shaded nodes. Thus, our latent variables are $\mathbf{Z} = \{D, I, G, S, L\}$, our data is $\mathbf{X} = \{H, J\}$, and a factorization for $p(\mathbf{z} \mid \mathbf{x})$ is desired.

The NaMI graph-inversion algorithm is traced in Table 1. Each step incrementally constructs two graphs: an *induced graph* $\mathcal{J}$ and a *stochastic inverse* $\mathcal{H}$. The induced graph is an undirected graph whose maximally connected subgraphs, or *cliques*, correspond to the scopes of the intermediate factors produced by simulating variable elimination. The stochastic inverse represents our eventual target which encodes the inverse dependency structure. It is constructed using information from the partially-constructed induced graph. Specifically, NaMI goes through the following steps for this example.

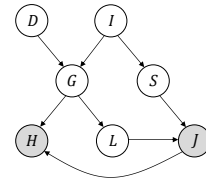

Figure 3: Example BN

**STEP 0**: The partial induced graph and stochastic inverse are initialized. The initial induced graph is formed by taking the directed graph for the forward model, $\mathcal{G}$, removing the directionality of the

Table 1: Tracing the NaMI algorithm on example from Figure 3. $S$ is the set of "frontier" variables that are considered for elimination, $v \in S$ the variable eliminated at each step chosen by the greedy min-fill heuristic, $\mathcal{J}$ the partially constructed induced graph *after* each step with black nodes indicating a eliminated variables, and $\mathcal{H}$ the partially constructed stochastic inverse.

| STEP | $S$ | $v$ | $\mathcal{J}$ | $\mathcal{H}$ | STEP | $S$ | $v$ | $\mathcal{J}$ | $\mathcal{H}$ |
|---|---|---|---|---|---|---|---|---|---|
| 0 | $\emptyset$ | $\emptyset$ | | | 3 | $G, S$ | $S$ | | |
| 1 | $D, I$ | $D$ | | | 4 | $G$ | $G$ | | |
| 2 | $I$ | $I$ | | | 5 | $L$ | $L$ | | |

edges, and adding additional edges between variables that share a child in $\mathcal{G}$—in this example, edges $D - I$, $S - L$ and $G - J$. This process is known as *moralization*. The stochastic inverse begins as disconnected variables, and edges are added to it at each step.

**STEP 1**: The frontier set of variables to consider for elimination, $S$, is initialized to the latent variables having no latent parents in $\mathcal{G}$, that is, $D, I$. To choose which variable to eliminate first, we apply the greedy min-fill heuristic, which is to choose the (possibly non-unique) variable that adds the fewest edges to the induced graph $\mathcal{J}$ in order to produce as compact an inverse as possible under the topological ordering. Specifically, noting that the cliques of $\mathcal{J}$ correspond to the scopes of intermediate factors during variable elimination, we want to avoid producing intermediate factors which would require us to add additional edges to $\mathcal{J}$, as doing so will in turn induce additional edges in $\mathcal{H}$ at future steps. For this example, if we were to eliminate $D$, that would produce an intermediate factor, $\psi_D(D, I, G)$, while if we were to eliminate $I$, that would produce an intermediate factor, $\psi_I(I, D, G, S)$. Choosing to eliminate would $I$ thus requires adding an edge $G$–$S$ to the induced graph, as there is no clique $I, D, G, S$ in the current state of $\mathcal{J}$. Conversely, eliminating $D$ does not require adding extra edges to $\mathcal{J}$ and so we choose to eliminate $D$.

The elimination of $D$ is simulated by marking its node in $\mathcal{J}$. The parents of $D$ in the inverse $\mathcal{H}$ are set to be its nonmarked neighbours in $\mathcal{J}$, that is, $I$ and $G$. $D$ is then removed from the frontier, and any non-observed children in $\mathcal{G}$ of $D$ whose parents have all been marked added to it—in this case, there are none as the only child of $D$, $G$, still has an unmarked parent $I$.

**STEP 2**: Variable $I$ is the sole member of the frontier and is chosen for elimination. The elimination of $I$ is simulated by marking its node in $\mathcal{J}$ *and* adding the additional edge $G$–$S$. This is required because elimination of $I$ requires the addition of a factor, $\psi_I(I, G, S)$, that is not currently present in $\mathcal{J}$. The parents of $I$ in the inverse $\mathcal{H}$ are set to be its nonmarked neighbours in $\mathcal{J}$, $G$ and $S$. $I$ is then removed from the frontier. Now, $G$ and $S$ are children of $I$, and both their parents $D$ and $I$ have been marked. Therefore, they are added to the frontier.

**STEP 3-5**: The process is continued until the end of the fifth step when all the latent variables, $D, I, S, G, L$, have been eliminated and the frontier is empty. At this point, $\mathcal{H}$ represents a factorization $p(\mathbf{z} \mid \mathbf{x})$, and we stop here as only a factorization for the posterior is required for amortized inference. Note, however, that it is possible to continue simulating steps of variable elimination on the observed variables to complete the factorization as $p(\mathbf{z} \mid \mathbf{x})p(\mathbf{x})$.

An important point to note is that NaMI's graph inversion can be run in one of two modes. The "topological mode," which we previously implicitly considered, simulates variable elimination in a topological ordering, producing an inverse that reverses the order of the random choices from the generative model. Conversely, NaMI's graph inversion can also be run in "reverse topological

mode," which simulates variable elimination in a reverse topological ordering, producing an inverse that preserves the order of random choices in the generative model. We will refer to these approaches as *forward-NaMI* and *reverse-NaMI* respectively in the rest of the paper. The rationale for these two modes is that, though they both produce minimally faithful inverses, one may be substantially more compact than the other, remembering that minimality only ensures a local optimum. For an arbitrary graph, it cannot be said in advance which ordering will produce the more compact inverse. However, as the cost of running the inversion algorithm is low, it is generally feasible to try and pick the one producing a better solution.

---

**Algorithm 1** NaMI Graph Inversion

---

1: **Input:** BN structure $\mathcal{G}$, latent variables $\mathcal{Z}$, TOPMODE?
2: $\mathcal{J} \leftarrow$ MORALIZE($\mathcal{G}$)
3: Set all vertices of $\mathcal{J}$ to be unmarked
4: $\mathcal{H} \leftarrow \{$VARIABLES($\mathcal{G}$), $\emptyset\}$, i.e. unconnected graph
5: UPSTREAM $\leftarrow$ "parent" if TOPMODE? else "child"
6: DOWNSTREAM $\leftarrow$ "child" if TOPMODE? else "parent"
7: $S \leftarrow$ all latent variables without UPSTREAM latents in $\mathcal{G}$
8: **while** $S \neq \emptyset$ **do**
9:    Select $v \in S$ according to min-fill criterion
10:   Add edges in $\mathcal{J}$ between unmarked neighbours of $v$
11:   Make unmarked neighbours of $v \in \mathcal{J}$, $v$'s parents in $\mathcal{H}$
12:   Mark $v$ and remove from $S$
13:   **for** unmarked latents DOWNSTREAM $u$ of $v$ in $\mathcal{G}$ **do**
14:     Add $u$ to $S$ if all its UPSTREAM latents in $\mathcal{G}$ are marked
15:   **end for**
16: **end while**
17: **return** $\mathcal{H}$

---

The general NaMI graph-reversal procedure is given in Algorithm 1. It is further backed up by the following formal demonstration of correctness, the proof for which is given in Appendix F.

**Theorem 1.** *The Natural Minimal I-Map Generator of Algorithm 1 produces inverse factorizations that are natural and minimally faithful.*

We further note that NaMI's graph reversal has a running time of order $O(nc)$ where $n$ is the number of latent variables in the graph and $c << n$ is the size of the largest clique in the induced graph. We consequently see that it can be run cheaply for practical problems: the computational cost of generating the inverse is generally dominated by that of training the resulting inference network itself. See Appendix F for more details.

### 2.3 Using the faithful inverse

Once we have obtained the faithful inverse structure $\mathcal{H}$, the next step is to use it to learn an inference network, $q_\psi(\mathbf{z} \mid \mathbf{x})$. For this, we use the factorization given by $\mathcal{H}$. Let $\tau$ denote the reverse of the order in which variables were selected for elimination by Line 9 in Algorithm 1, such that $\tau$ is a permutation of $1, \ldots, n$ and $\tau(n)$ is the first variable eliminated. $\mathcal{H}$ encodes the factorization

$$q_\psi(\mathbf{z} \mid \mathbf{x}) = \prod_{i=1}^{n} q_i(z_{\tau(i)} \mid \text{Pa}_{\mathcal{H}}(z_{\tau(i)})) \tag{1}$$

where $\text{Pa}_{\mathcal{H}}(z_{\tau(i)}) \subseteq \{\mathbf{x}, z_{\tau(1)}, \ldots, z_{\tau(i-1)}\}$ indicates the parents of $z_{\tau(i)}$ in $\mathcal{H}$. For each factor $q_i$, we must decide both the class of distributions for $z_{\tau(i)} \mid \text{Pa}_{\mathcal{H}}(z_{\tau(i)})$, and how the parameters for that class are calculated. Once learned, we can both sample from, and evaluate the density of, the inference network for a given dataset by considering each factor in turn.

The most natural choice for the class of distributions for each factor is to use the same distribution family as the corresponding variable in the generative model, such that the supports of these distributions match. For instance, continuing the example from Figure 3, if $D \sim N(0, 1)$ in the generative model, then a normal distribution would also be used for $D \mid I, G$ in the inference network. To establish the mapping from data to the parameters to this distribution, we train neural networks using stochastic gradient ascent methods. For instance, we could set $D \mid \{I = i, G = g\} \sim N(\mu_\varphi(i, g), \sigma_\varphi(i, g))$, where $\mu_\varphi$ and $\sigma_\varphi$ are two densely connected feedforward networks, with learnable parameters $\varphi$. In general, it will be important to choose architectures which well match the problem at hand. For example, when perceptual inputs such as images and language are present in the conditioning variables, it is advantageous to first embed them to a lower-dimensional representation using, for example, convolutional neural networks.

Matching the distribution families in the inference network and generative model, whilst a simple and often adequate approximation, can be suboptimal. For example, suppose that for a normally distributed variable in the generative model, the true conditional distribution in the posterior for that variable is multimodal. In this case, using a (single mode) normal factor in the inference network would not suffice. One could straightforwardly instead use, for example, either a mixture of Gaussians, or, normalizing flows (Rezende & Mohamed, 2015; Kingma et al., 2016), to parametrize each inference network factor in order to improve expressivity, at the cost of additional implementational

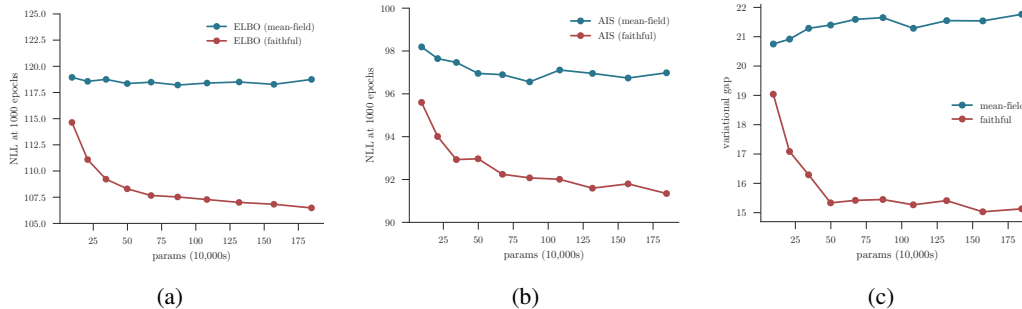

Figure 4: Results for the relaxed Bernoulli VAE with 30 latent units, compared after 1000 epochs of learning the: (a) negative ELBO, and (b) negative AIS estimates, varying inference network factorizations and capacities (total number of parameters); (c) An estimate of the variational gap, that is, the difference between marginal log-likelihood and the ELBO.

complexity. In particular, if one were to use a provably universal density estimator to parameterize each inference network factor, such as that introduced in Huang et al. (2018), the resulting NaMI inverse would constitute a universal density estimator of the true posterior.

After the inference network has been parametrized, it can be trained in number of different ways, depending on the final use case of the network. For example, in the context of amortized stochastic variational inference (SVI) methods such as VAEs (Kingma & Welling, 2014; Rezende et al., 2014), the model $p_\theta(\mathbf{x}, \mathbf{z})$ is learned along with the inference network $q_\psi(\mathbf{z} \mid \mathbf{x})$ by optimizing a lower bound on the marginal loglikelihood of the data, $\mathcal{L}_{ELBO} = \mathbb{E}_{q_\psi(\mathbf{z}|\mathbf{x})} [\ln p_\theta(\mathbf{x}, \mathbf{z}) - \ln q_\psi(\mathbf{z} \mid \mathbf{x})]$. Stochastic gradient ascent can then be used to optimize $\mathcal{L}_{ELBO}$ in the same way a standard VAE, simulating from $q_\psi(z|x)$ by considering each factor in turn and using reparameterization (Kingma & Welling, 2014) when the individual factors permit doing so.

A distinct training approach is provided when the model $p(\mathbf{x}, \mathbf{z})$ is fixed (Papamakarios & Murray, 2015). Here a proposal is learnt for either importance sampling (Le et al., 2017) or sequential Monte Carlo (Paige & Wood, 2016) by using stochastic gradient ascent to minimize the reverse KL-divergence between the inference network $q_\psi(\mathbf{z} \mid \mathbf{x})$ and the true posterior $p(\mathbf{z} \mid \mathbf{x})$. Up to a constant, the objective is given by $\mathcal{L}_{IC} = \mathbb{E}_{p(\mathbf{x}, \mathbf{z})} [-\ln q_\psi(\mathbf{z} \mid \mathbf{x})]$.

Using a minimally faithful inverse structure typically improves the best inference network attainable and the finite time training performance for both these settings, compared with previous naive approaches. In the VAE setting, this can further have a knock-on effect on the quality of the learned model $p_\theta(\mathbf{x}, \mathbf{z})$, both because a better inference network will give lower variance updates of the generative network (Rainforth et al., 2018) and because restrictions in the expressiveness of the inference network lead to similar restrictions in the generative network (Cremer et al., 2017, 2018).

In deep generative models, the BNs may be much larger than the examples shown here. However, typically at the macro-level, where we collapse each vector to a single node, they are quite simple. When we invert this type of collapsed graph, we must do so with the understanding that the distribution over a vector-valued node in the inverse must express dependencies between all its elements in order for the inference network to be faithful.

## 3 Experiments

We now consider the empirical impact of using NaMI compared with previous approaches. In §3.1, we highlight the importance of using a faithful inverse in the VAE context, demonstrating that doing so results in a tighter variational bound and a higher log-likelihood. In §3.2, we use NaMI in the fixed-model setting. Here our results demonstrate the importance of using both a faithful and minimal inverse on the efficiency of the learned inference network. Low-level details on the experimental setups can be found in Appendix D and an implementation at `https://git.io/fxVQu`.

### 3.1 Relaxed Bernoulli VAEs

Prior work has shown that more expressive inference networks give an improvement in amortized SVI on sigmoid belief networks and standard VAEs, relative to using the mean-field approximation (Uria et al., 2016; Maaløe et al., 2016; Rezende & Mohamed, 2015; Kingma et al., 2016). Krishnan et al. (2017) report similar results when using more expressive inverses in deep linear-chain state-space models. It is straightforward to see that any minimally faithful inverse for the standard VAE

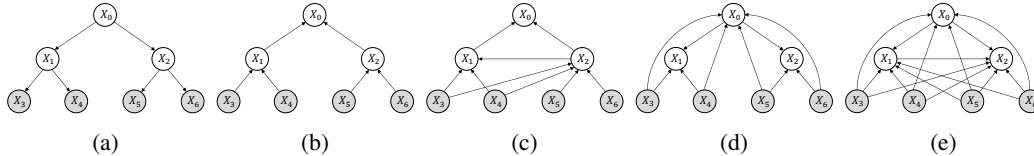

(a)  (b)  (c)  (d)  (e)

Figure 5: (a) BN structure for a binary tree with $d = 3$; (b) Stuhlmüller's heuristic inverse; (c) Natural minimally faithful inverse produced by NaMI in topological mode; (d) Most compact inverse when $d > 3$, given by running NaMI in reverse topological mode; (e) Fully connected inverse.

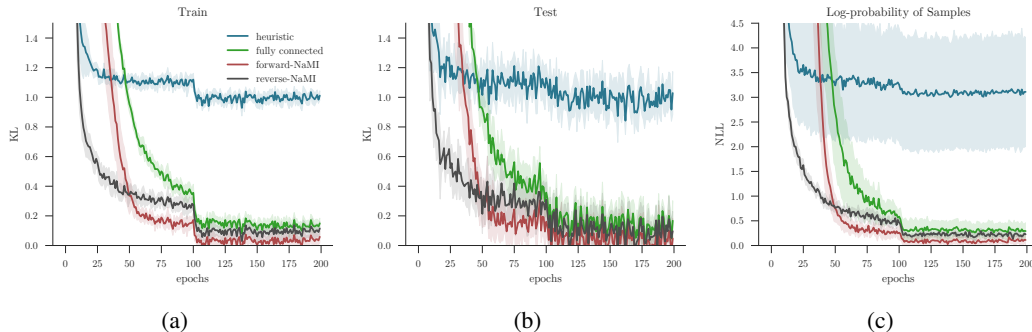

(a)  (b)  (c)

Figure 6: Results for binary tree Gaussian BNs with depth $d = 5$, comparing inference network factorizations in the compiled inference setting. The KL divergence from the analytical posterior estimated to the inference network on the training and test sets are shown in (a) and (b) respectively. (c) shows the average negative log-likelihood of inference network samples under the analytical posterior, conditioning on five held-out data sets. The results are averaged over 10 runs and 0.75 standard deviations indicated. The drop at 100 epochs is due to decimating the learning rate.

framework (Kingma & Welling, 2014) has a fully connected clique over the latent variables so that the inference network can take account of the explaining-away effects between the latent variables in the generative model. As such, both forward-NaMI and backward-NaMI produce the same inverse.

The relaxed Bernoulli VAE (Maddison et al., 2017b; Jang et al., 2017) is a VAE variation that replaces both the prior on the latents and the distribution over the latents given the observations with the relaxed Bernoulli distribution (also known as the Concrete distribution). It can also be understood as a "deep" continuous relaxation of sigmoid belief networks.

We learn a relaxed Bernoulli VAE with 30 latent variables on MNIST, comparing a faithful inference network (parametrized with MADE (Germain et al., 2015)) to the mean-field approximation, after 1000 epochs of learning for ten different sizes of inference network, keeping the size of the generative network fixed. We note that the mean-field inference network has the same structure as the heuristic one that reverses the edges from the generative model. A tight bound on the marginal likelihood is estimated with annealed importance sampling (AIS) (Neal, 1998; Wu et al., 2017).

The results shown in Figure 4 indicate that using a faithful inverse on this model produces a significant improvement in learning over the mean-field inverse. Note that the x-axis indicates the number of parameters in the inference network. We observe that for *every* capacity level, the faithful inference network has a lower negative ELBO and AIS estimate than that of the mean-field inference network. In Figure 4c, the variational gap is observed to decrease (or rather, the variational bound tightens) for the faithful inverse as its capacity is increased, whereas it increases for the mean-field inverse. This example illustrates the inadequacy of the mean-field approximation in certain classes of models, in that it can result in significantly underutilizing the capacity of the model.

## 3.2 Binary-tree Gaussian BNs

Gaussian BNs are a class of models in which the conditional distribution of each variable is normally distributed, with a fixed variance and a mean that is a fixed linear combination of its parents plus an offset. We consider here Gaussian BNs with a binary-tree structured graph and observed leaves (see Figure 5a for the case of depth, $d = 3$). In this class of models, the exact posterior can be calculated analytically (Koller & Friedman, 2009, §7.2) and so it forms a convenient test-bed for performance.

The heuristic inverses simply invert the edges of the graph (Figure 5b), whereas a natural minimally faithful inverse requires extra edges between subtrees (e.g. Figure 5c) to account for the influence one

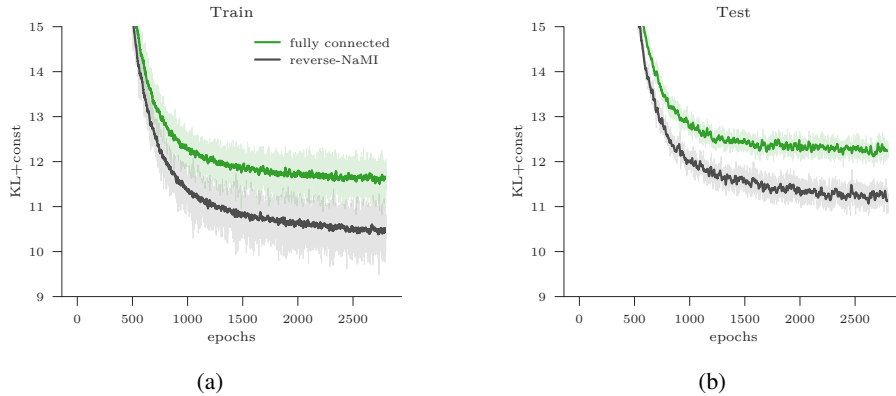

<div style="text-align:center">(a) &emsp;&emsp;&emsp;&emsp;&emsp;&emsp;&emsp;&emsp;&emsp;&emsp;&emsp; (b)</div>

Figure 7: Convergence of reverse KL divergence (used as the training objective) for Bayesian GMM for $K = 3$ clusters and $N = 200$ data points, comparing inference networks with a fixed generative model. The shaded regions indicate 1 standard error in the estimation.

node can have on others through its parent. For this problem, it turns out that running reverse-NaMI (Figure 5d) produces a more compact inverse than forward-NaMI. This, in fact, turns out to be the most compact possible I-map for any $d > 3$. Nonetheless, all three inversion methods have significantly fewer edges than the fully connected inverse (Figure 5e).

The model is fixed and the inference network is learnt from samples from the generative model, minimizing the "reverse" KL-divergence, namely that from the posterior to the inference network $\mathrm{KL}(p_\theta(\mathbf{z}|\mathbf{x})||q_\psi(\mathbf{z}|\mathbf{x}))$, as per (Paige & Wood, 2016). We compared learning across the inverses produced by using Stuhlmüller's heuristic, forward-NaMI, reverse-NaMI, and taking the fully connected inverse. The fully connected inference network was parametrized using MADE (Germain et al., 2015), and the forward-NaMI one with a novel MADE variant that modifies the masking matrix to exactly capture the tree-structured dependencies (see Appendix E.2). As the same MADE approaches cannot be used for heuristic and reverse-NaMI inference networks, these were instead parametrized with a separate neural network for each variable's density function. The inference network sizes were kept constant across approaches.

Results are given in Figure 6 for depth $d = 5$ averaging over 10 runs. Figures 6a and 6b show an estimate of $\mathrm{KL}(p_\theta(\mathbf{z}|\mathbf{x})||q_\psi(\mathbf{z}|\mathbf{x}))$ using the train and test sets respectively. From this, we observe that it is necessary to model at least the edges in an I-map for the inference network to be able to recover the posterior, and convergence is faster with fewer edges in the inference network. Despite the more compact reverse-NaMI inverse converging faster than the forward-NaMI one, the latter seems to converges to a better final solution. This may be because the MADE approach could not be used for the reverse-NaMI inverse, but this is a subject for future investigation nonetheless.

Figure 6c shows the average negative log-likelihood of 200 samples from the inference networks evaluated on the analytical posterior, conditioning on five fixed datasets sampled from the generative model not seen during learning. It is thus a measure of how successful inference amortiziation has been. All three faithful inference networks have significantly lower variance over runs compared to the unfaithful inference network produced by Stuhlmüller's algorithm.

We also observed during other experimentation that if one were to decrease the capacity of all methods, learning remains stable in the natural minimally faithful inverse at a threshold where it becomes unstable in the fully connected case and in Stuhlmüller's inverse.

## 3.3 Gaussian Mixture Models

Gaussian mixture models (GMMs) are a clustering model where the data $\mathbf{x} = \{x_1, x_2, \ldots, x_N\}$ is assumed to have been generated from one of $K$ clusters, each of which has a Gaussian distribution with parameters $\{\mu_j, \Sigma_j\}$, $j = 1, 2, \ldots, K$. Each datum, $x_i$ is associated with a corresponding index, $z_i \in \{1, \ldots, K\}$ that gives the identity of that datum's cluster. The indices, $\mathbf{z}' = \{z_i\}$ are drawn i.i.d. from a categorical distribution with parameter $\phi$. Prior distributions are placed on $\theta = \{\mu_1, \Sigma_1, \ldots, \mu_K, \Sigma_K\}$ and $\phi$, so that the latent variables are $\mathbf{z} = \{\mathbf{z}', \theta, \phi\}$. The goal of inference is then to determine the posterior $p(\mathbf{z} \mid \mathbf{x})$, or some statistic of it.

As per the previous experiment, this falls into the fixed-model setting. We factor the fully-connected inverse as, $q(\theta|x)q(\phi|\theta, \mathbf{x})q(\mathbf{z}'|\phi, \theta, \mathbf{x})$. It turns out that applying reverse-NaMI de-

couples the dependence between the indices, $\mathbf{z}'$, and produces a much more compact factorization, $q(\theta|\mathbf{x}, \phi) \prod_i^N q(z_i|x_i, \phi, \theta)q(\phi|\mathbf{x})$, than either the fully-connected or forward-NaMI inverses for this model. The inverse structure produced by Stuhlmüller's heuristic algorithm is very similar to the reverse-NaMI structure for this problem and is omitted.

We train our amortization artifact over datasets with $N = 200$ samples and $K = 3$ clusters. The inference network terms with distributions over vectors were parametrized by MADE, and we compare the results for the fully-connected and reverse-NaMI inverses. We hold the neural network capacities constant across methods and average over 10 runs, the results for which are shown in Figure 7. We see that learning is faster for the minimally faithful reverse-NaMI method, relative to the fully-connected inverse, and converges to a better solution, in agreement with the other experiments.

### 3.4 Minimal and Non-minimal Faithful Inverses

To further examine the hypothesis that a non-minimal faithful inverse has slower learning and converges to a worse solution relative to a minimal one, we performed the setup of Experiment 3.2 with depth d = 4, comparing the forward-NaMI network to two additional networks that added 12 and 16 connections to forward-NaMI (holding the total capacity fixed).

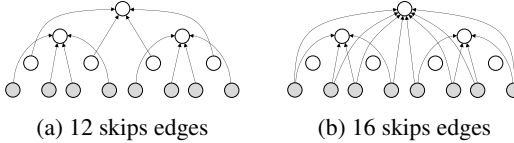

(a) 12 skips edges     (b) 16 skips edges

Figure 8: Additional edges over forward-NaMI.

The additional edges are shown in Figure 8. Note the regular forward-NaMI edges are omitted for visual clarity.

Figure 9 shows the average negative log likelihood (NLL) under the true posterior for samples generated by the inference network, based on 5 datasets not seen during training. It appears that the more edges are added beyond minimality, the slower is the initial learning and convergence is to a worse solution.

To further explain why minimality is crucial, we note that adding additional edges beyond minimality means that there will be factors that condition on variables whose probabilistic influence is blocked by the other variables. This effectively adds an input of random noise into these factors, which is why we then see slower learning and convergence to a worse solution.

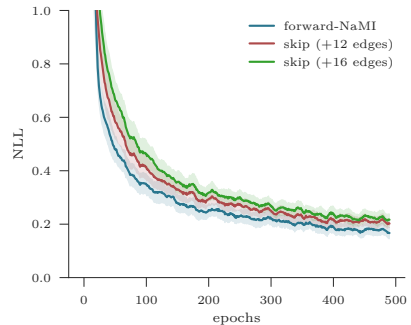

Figure 9: Average NLL of inference network samples under analytical posterior.

## 4 Discussion

We have presented NaMI, a tractable framework that, given the BN structure for a generative model, produces a natural factorization for its inverse that is a minimal I-map for the model. We have argued that this should be used to guide the design of the coarse-grain structure of the inference network in amortized inference. Having empirically analyzed the implications of using NaMI, we find that it learns better inference networks than previous heuristic approaches. We further found that, in the context of VAEs, improved inference networks have a knock-on effect on the generative network, improving the generative networks as well.

Our framework opens new possibilities for learning structured deep generative models that combine traditional Bayesian modeling by probabilistic graphical models with deep neural networks. This allows us to leverage our typically strong knowledge of which variables effect which others, while not overly relying on our weak knowledge of the exact functional form these relationships take.

To see this, note that if we forgo the niceties of making mean-field assumptions, we can impose arbitrary structure on a generative model simply by controlling its parameterization. The only requirement on the generative network to evaluate the ELBO is that we can evaluate the network density at a given input. Recent advances in normalizing flows (Huang et al., 2018; Chen et al., 2018) mean it is possible to construct flexible and general purpose distributions that satisfy this requirement and are amenable to application of dependency constraints from our graphical model. This obviates the need to make assumptions such as conjugacy as done by, for example, Johnson et al. (2016).

NaMI provides a critical component to constructing such a framework, as it allows one to ensure that the inference network respects the structural assumptions imposed on the generative network, without which a tight variational bound cannot be achieved.

## Acknowledgments

We would like to thank (in alphabetical order) Rob Cornish, Rahul Krishnan, Brooks Paige, and Hongseok Yang for their thoughtful help and suggestions.

SW and AG gratefully acknowledge support from the EPSRC AIMS CDT through grant EP/L015987/2. RZ acknowledges support under DARPA D3M, under Cooperative Agreement FA8750-17-2-0093. NS was supported by EPSRC/MURI grant EP/N019474/1. TR and YWT are supported in part by the European Research Council under the European Union's Seventh Framework Programme (FP7/2007–2013) / ERC grant agreement no. 617071. TR further acknowledges support of the ERC StG IDIU. FW was supported by The Alan Turing Institute under the EPSRC grant EP/N510129/1, DARPA PPAML through the U.S. AFRL under Cooperative Agreement FA8750-14-2-0006, an Intel Big Data Center grant, and DARPA D3M, under Cooperative Agreement FA8750-17-2-0093.

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
