[Supplementary Material]

# Faithful Inversion of Generative Models for Effective Amortized Inference: Supplementary Material

**Stefan Webb**[*]
University of Oxford

**Adam Goliński**
University of Oxford

**Robert Zinkov**
UBC

**N. Siddharth**
University of Oxford

**Tom Rainforth**
University of Oxford

**Yee Whye Teh**
University of Oxford

**Frank Wood**
UBC

## A  Probabilistic Graphical Models

This summary is based on Koller & Friedman (2009).

### A.1  Bayesian networks and representation

Any probability distribution implicitly represents certain independence relationships between its variables via its factorization. These are of interest because they can be exploited to both compactly represent distributions and to reduce the cost of inference. The set of such relationships is defined as:

**Definition 1.** *Let $p$ be a distribution defined over $\mathcal{X}$. We define $\mathcal{I}(p)$ to be the set of* independence assertions *of the form* $(\mathbf{X} \perp \mathbf{Y} \mid \mathbf{Z})$ *that hold in $p$, where* $\mathbf{X}, \mathbf{Y}, \mathbf{Z} \subseteq \mathcal{X}$.

The framework of probabilistic graphical models is used for representing and reasoning about a wide class of probability distributions by making these independence assertions explicit. Distributions are represented as the product of factors over subsets of the model variables. Associated with the factorization is a graph, wherein the nodes are the random variables of the model, and the edges express the distribution's independence assertions.

Bayesian networks (BNs) are a class of probabilistic graphical models that use a directed acyclic graph. We refer to the graph alone as the BN structure, whereas the BN itself comprises, in addition, a representation for each factor. In a BN, each variable has a conditional distribution that only depends on its parents in the graph. For example, in Figure 2a the distribution factors as $p(a)p(b|a)p(c|a)p(d|b)p(e|c)$.

Formally, the semantics of the BN structure are that it encodes the local independencies:

**Definition 2.** *A Bayesian network structure $\mathcal{G}$ encodes the* local independencies *$\mathcal{I}_l(\mathcal{G})$, namely, those of the form $X_i \perp \text{NonDescendants}_{X_i} \mid \text{Pa}_{X_i}^{\mathcal{G}}$ for each $X_i \in \mathcal{G}$, where $\text{Pa}_{X_i}^{\mathcal{G}}$ denotes the parents of $X_i$ in $\mathcal{G}$.*

It turns out that there are additional independencies that can be read off $\mathcal{G}$ aside from the local ones, that hold for every $p$ that factorizes over $\mathcal{G}$, and these are identified by the concept of *d-separation*.

We relate the conditional independencies encoded in a graph, such as a BN structure, to a corresponding distribution by the concept of an independency map, or *I-map*:

**Definition 3.** *Let $\mathcal{K}$ be any graph object associated with a set of independencies $\mathcal{I}(\mathcal{K})$. We say that $\mathcal{K}$ is an* I-map *for a distribution $p$ if $\mathcal{I}(\mathcal{K}) \subseteq \mathcal{I}(p)$.*

In our case, a BN structure $\mathcal{G}$ is an I-map for $p$ if $\mathcal{I}_l(\mathcal{G}) \subseteq \mathcal{I}(p)$. This means that $\mathcal{G}$ may not encode all the independencies in $p$, *but it does not mislead us by encoding independencies not present in $p$.* For this reason, we will interchangeability use the expression, "$\mathcal{G}$ is faithful to $p$."

---

[*]Correspondence to info@stefanwebb.me

It can be proven that a BN structure $\mathcal{G}$ is an I-map for a distribution $p$ if and only if $p$ is representable as a set of conditional probability distributions (also referred to as model factors), factoring according to $\mathcal{G}$, that is,

$$P(\mathcal{X}) = \prod_{X_i \in \mathcal{X}} P(X_i \mid \mathrm{Pa}_{X_i}^{\mathcal{G}}).$$

Therefore, we can use the graph as a means of revealing the structure in a distribution.

## A.2 D-separation

We give a heuristic explanation of d-separation by examining the opposite question of, roughly speaking, when can probabilistic influence flow from one variable to another.

In paths in $\mathcal{G}$ with three variables that form,

- a causal trail, $X \to Z \to Y$,
- an evidential trail, $X \leftarrow Z \leftarrow Y$, or,
- a common cause, $X \leftarrow Z \to Y$,

knowledge of $X$ is informative about $Y$ when $Z$ is not observed, and observing $Z$ blocks this flow of information. For example, suppose $X$ is the coherence of a course, $Z$ its difficulty, and $Y$ the grade a student receives. Further, suppose there is a causal trail $X \to Z \to Y$ in the graph and no other trail between $X$ and $Y$. If we observe that the course is taught coherently, this will inform our beliefs about its difficulty, which will in turn change our beliefs about the student's grade. On the other hand, if we observe that it is a difficult course, the coherency of the course will not effect our beliefs about the student's grade as it can only do so indirectly via the difficulty variable.

Conversely, for a common effect motif, $X \to Z \leftarrow Y$, also known as a *v-structure*, there is an "explaining away" effect, whereby if we observe $Z$ (or a descendent of $Z$), then knowledge of $X$ *is* informative about $Y$. For example, if $X$ if the difficulty of an exam, $Z$ is a student's result, and $Y$ is his aptitude, then if we observe a poor result and that the exam is hard, we can attribute the result to the difficulty of the exam, and lessen our belief that the student is incapable.

This heuristic reasoning generalizes to longer trails in the concept of an *active trail*,

**Definition 4.** *Let $\mathcal{G}$ be a BN structure and $X_1 \rightleftharpoons \cdots \rightleftharpoons X_n$ a trail in $\mathcal{G}$. Let $\mathbf{Z}$ be a subset of observed variables. The trail $X_1 \rightleftharpoons \cdots \rightleftharpoons X_n$ is* active *given $\mathbf{Z}$ if,*

- *Whenever we have a v-structure $X_{i-1} \to X_i \leftarrow X_{i+1}$, then $X_i$ or one of its descendants are in $\mathbf{Z}$;*

- *No other node along the trail is in $\mathbf{Z}$.*

Those subsets of variables, conditioned on another set, are said to be d-separated if an active trail does not exist between them. Formally:

**Definition 5.** *Let $\mathbf{X}, \mathbf{Y}, \mathbf{Z}$ be three sets of nodes in $\mathcal{G}$.*

*We say that $\mathbf{X}$ and $\mathbf{Y}$ are d-separated given $\mathbf{Z}$, denoted* d-sep$_{\mathcal{G}}(\mathbf{X}; \mathbf{Y} \mid \mathbf{Z})$, *if there is no active trail between any node $X \in \mathbf{X}$ and $Y \in \mathbf{Y}$ given $\mathbf{Z}$. We use $\mathcal{I}(\mathcal{G})$ to denote the set of independencies that correspond to d-separation,*

$$\mathcal{I}(\mathcal{G}) = \{(\mathbf{X} \perp \mathbf{Y} \mid \mathbf{Z}) \mid \text{d-sep}_{\mathcal{G}}(\mathbf{X}; \mathbf{Y} \mid \mathbf{Z})\}.$$

D-separation is sound in the sense that if $\mathbf{X}$ and $\mathbf{Y}$ are d-separated given $\mathbf{Z}$ in a graph $\mathcal{G}$, then $\mathbf{X} \perp \mathbf{Y} \mid \mathbf{Z}$ holds in all distributions $p$ that factorize according to $\mathcal{G}$ (Koller & Friedman, 2009, Theorem 3.3).

A certain converse statement also holds for the completeness of d-separation. If $\mathbf{X}$ and $\mathbf{Y}$ are not d-separated given $\mathbf{Z}$ in a graph $\mathcal{G}$, then $\mathbf{X} \perp \mathbf{Y} \mid \mathbf{Z}$ does not hold for almost all (in a measure theoretic sense) distributions $p$ that factorize according to $\mathcal{G}$ (Koller & Friedman, 2009, Theorem 3.5). So, for all practical purposes one may assume $\mathcal{I}(\mathcal{G}) = \mathcal{I}(p)$.

Figure 1: (a) BN structure for "Extended Student" example; (b) the induced graph corresponding to elimination ordering $D, I, H, G, S, L$; (c) the corresponding clique tree; (d) the clique tree corresponding to elimination ordering $D, I, S, G, L, J, H$.

## A.3 Exact inference by variable elimination

Variable elimination is an algorithm for performing exact inference in graphical models which have the property that summation of variables in the model factors is tractable—typically ones with discrete finite-valued factors. From a higher perspective, it works by using the observation that we can exchange the order of the summation of the model variables and the multiplication of the model factors based on their scope, i.e. what variables they take as inputs. Doing so can greatly reduce the complexity of summation, or rather inference, if the variable ordering is carefully chosen.

Consider the BN structure from Figure 1a and suppose the task is to compute $P(J)$. Simply multiplying all the factors together, then summing out $\mathcal{X} \setminus \{J\}$,

$$P(J) = \sum_{\mathcal{X} \setminus \{J\}} \prod_{X \in \mathcal{X}} \phi_X,$$

would not be an efficient means to do so. Rather, we ought to exploit the structure in the model, and perform summation on factors with smaller scope. Suppose also, that we perform the summation, or variable elimination, in the ordering $D, I, H, G, S, L$. To sum out $D$, we can pull out all factors that do not contain $D$ in their scope. First we multiply the factors depending on $D$ together,

$$\psi_1(D) = \phi_D(D)\phi_G(G, I, D),$$

then sum out $D$,

$$\tau_1(G, I) = \sum_D \psi_1,$$

to produce a new intermediate factor that is used in subsequent computations.

Similarly, to sum out $I$,

$$\psi_2(G, I, S) = \tau_1(G, I)\phi_I(I)\phi_S(S, I),$$
$$\tau_2(G, I) = \sum_I \psi_2.$$

continuing this process to eliminate the remaining variables. As each intermediate factor, $\psi_i$, has a scope much narrower than the full variables set, $\mathcal{X}$, exact inference is made tractable.

## A.4 Induced graphs

The computational cost of an application of variable elimination, which depends on the size of the scope of the largest intermediate factor, can be captured in an undirected graph known as the *induced graph*. It is defined as follows:

**Definition 6.** *Let $\Phi$ be a set of factors over $\mathcal{X} = \{X_1, \ldots, X_n\}$, and $\prec$ be an elimination ordering for some subset $\mathcal{X} \subseteq \mathcal{X}$. The induced graph $\mathcal{I}_{\Phi, \prec}$ is an undirected graph over $\mathcal{X}$, where $X_i$ and $X_j$ are connected by an edge if they both appear in some intermediate factor $\phi$ generated by the variable elimination algorithm using $\prec$ as an elimination ordering.*

The induced graph for our previous example is given in Figure 1b. We see that it has cliques, or maximally connected subgraphs, for the subsets $\{D, I, G\}$, $\{I, S, G\}$, $\{G, J, S, L\}$, and $\{G, H, J\}$, which correspond to the scopes of some intermediate factor, $\psi_i$, in the computation.

We can form the induced graph for a given run of variable elimination on $\mathcal{G}$ as follows. First, we "moralize" $\mathcal{G}$ by connecting all its parents and removing the directionality of the edges. This induces an edge between $X_i$ and $X_j$ if they appear in the scope of a model factor $\phi \in \Phi$ before variable elimination. During variable elimination, after we have calculated the scope of each intermediate factor, we add additional edges to the graph, indicated in our figures with dotted edges, so that the scope of each intermediate factor, $\psi_i$, is maximally connected. For instance, in our example, when eliminating $I$, a factor $\psi_3(G, I, S)$ occurs, so we must add the additional edge $G - S$. A good variable elimination ordering will add as few additional edges so that the scope of the intermediate factors is constrained.

## A.5 Clique trees

Another way to understand the variable elimination algorithm is as an algorithm that passes messages over a tree structure known as a clique tree. Continuing our running "Student" example, the clique tree corresponding to the variable elimination ordering $D, I, H, G, S, L$ is given in Figure 1c. We refer to the nodes in the tree as the cliques, which are subsets of the model variables corresponding to the scopes of the intermediate factors, $\{\psi_i\}$. Each model factor, $\phi_i$, is associated to a node in the graph, for example, $\phi_D(D)$, $\phi_G(D, I, G)$, and $\phi_I(I)$ are associated with the node "$D, I, G$," and $\phi_S(I, S)$ is associated with "$I, S, G$."

The messages, $\{\tau_i\}$, are formed by multiplying together all the factors associated with a node and its incoming messages, and summing out the variables not in the intersection of the node and its downstream neighbour. The intersections of the node scopes are indicated above each edge and are known as the sepsets. The tree is undirected, although we have indicated the directionality of message passing with arrows above each edge.

Formally, a clique tree is defined as follows:

**Definition 7.** *A clique tree $\mathcal{U}$ for a set of factors $\Phi$ over $\mathcal{X}$ is an undirected graph, each of whose nodes $i$ is associated with a subset $\mathbf{C}_i \subset \mathcal{X}$. A clique tree must be family-preserving—each factor $\phi \in \Phi$ must be associated with a clique $\mathbf{C}_i$ such that $\text{scope}[\phi] \subseteq \mathbf{C}_i$. Each edge between a pair of cliques $\mathbf{C}_i$ and $\mathbf{C}_j$ is associated with a sepset $\mathbf{S}_{i,j} \subseteq \mathbf{C}_i \cap \mathbf{C}_j$. Also, it must hold that whenever there is a variable $X$ such that $X \in \mathbf{C}_i$ and $X \in \mathbf{C}_j$, then $X$ is also in every clique in the (unique) path in $\mathcal{T}$ between $\mathbf{C}_i$ and $\mathbf{C}_j$.*

An important property of clique trees, known as the *sepset property*, is the following: all variables upstream of a clique are conditionally independent of those downstream, conditioned on the corresponding sepset, and the sepset is the minimal set for which this holds (Koller & Friedman, 2009, Theorem 10.2). In this way, the sepset "separates" upstream and downstream variables. Property 1 in B.4 is equivalent to the sepset property—our definition of "upstream/downstream" coincides in induced graphs and clique trees, and the sepsets are seen to correspond to the downstream neighbours of a variable. Compare the induced graph of §2.2 with its corresponding clique tree in Figure 1d.

## A.6 Exact inverses

Is it possible in general for a stochastic inverse $\mathcal{H}$ to perfectly capture the independencies in $\mathcal{G}$ so that $\mathcal{I}(\mathcal{H}) = \mathcal{I}(\mathcal{G})$? The answer is given in the negative by the following theorem and associated definitions (Koller & Friedman, 2009, Theorem 3.8):

**Definition 8.** *The* skeleton *of a BN structure $\mathcal{G}$ over $\mathcal{X}$ is an undirected graph over $\mathcal{X}$ that contains an edge $\{X, Y\}$ for every edge $(X, Y)$ in $\mathcal{G}$.*

**Definition 9.** *A v-structure $X \rightarrow Y \leftarrow Z$ is an* immorality *if there is no direct edge between $X$ and $Y$.*

**Theorem 1.** *Let $\mathcal{G}$ and $\mathcal{H}$ be two graphs over $\mathcal{X}$. Then $\mathcal{G}$ and $\mathcal{H}$ have the same skeleton and the same set of immoralities if and only $\mathcal{I}(\mathcal{H}) = \mathcal{I}(\mathcal{G})$.*

In general, immoralities in $\mathcal{G}$ are destroyed in $\mathcal{H}$, as both heuristic and faithful inversion methods may reverse edges in v-structures or add a direct edge between their parents.

Figure 2: (a,d) Two simple BN structures for a generative model, (b,e) The corresponding inverse BN structures formed by Stuhlmüller's Algorithm, (c,f) The inverse BN structure formed by our algorithm. This demonstrates how Stuhlmüller's Algorithm can miss many edges and longer-term dependencies.

## B    Restrictions on orderings

So far, we have been simulating variable elimination on the latent variables in the model, stopping at the observed ones. In special cases, we may wish to further restrict the variable elimination ordering within the non-observed variables. For instance, the semi-supervised variational objective of Kingma et al. (2014) requires a factorization $q(\mathbf{z}, \mathbf{y} \mid \mathbf{x}) = q(\mathbf{z} \mid \mathbf{x}, \mathbf{y}) q(\mathbf{y} \mid \mathbf{x})$, where $\mathbf{y}$ are the semi-observed variables. In this case we should eliminate all $\mathbf{z}$ before eliminating $\mathbf{y}$. Algorithm 1 can be suitably modified to accommodate this by running Lines 6–17, replacing "latents" and "latent variables" with $z \in \mathbf{z}$, and repeating Lines 6–16 replacing those terms with $y \in \mathbf{y}$. In a time series model, we may wish to eliminate the latent variables in their time ordering, $\mathbf{z_1}, \ldots, \mathbf{z_T}$, and can repeat Lines 6–16 $T$ times, replacing those terms with $z \in \mathbf{z_i}$ in turn.

## C    Counterexamples to Stuhlmüller's heuristic inversion

Stuhlmüller et al. (2013) give an algorithm for forming a "heuristic inverse," $\mathcal{H}$, of a BN structure, $\mathcal{G}$.

First, let us define the concept of a Markov Blanket in a BN:

**Definition 10.** *Let $\mathcal{G}$ be a BN structure over $\mathcal{X}$. Then, the Markov blanket of $X \in \mathcal{X}$ in $\mathcal{G}$, Markov$_{\mathcal{G}}(X)$, is the minimal set of variables, $\mathbf{Z}$, that when conditioned on, make $X$ independent of $\mathcal{X} \setminus X$—that is, the set of parents, child, and parents of children of $X$.*

It is necessary to condition on the parents of a variable's children, because conditioning on its children may activate v-structures, and so we need to condition on the children's parents to block these paths.

Stuhlmüller's algorithm works by visiting the variables of $\mathcal{G}$ in a reverse topological ordering, $Y_1, \ldots, Y_n$ (where $Y_i$ is equal to some observed $X_j$ or latent $Z_k$ depending on the structure of the graph and the ordering). The graph $\mathcal{H}$ is produced by setting the parents of $Y_i$ to be the intersection of $Y_1, \ldots, Y_{i-1}$ and that node's Markov blanket in $\mathcal{G}$, excluding latent parents for observed nodes. The procedure is equivalent to reversing the edges in $\mathcal{G}$, adding extra edges to fully connect all the parents of a node in $\mathcal{G}$, and removing edges from latent nodes into observed ones. This produces the desired factorization $q(\mathbf{x} \mid \mathbf{z}) q(\mathbf{z})$.

Paige & Wood (2016) claim that a heuristic inverse structure $\mathcal{H}$ is an I-map for $\mathcal{G}$, or equivalently, by the almost-everywhere completeness of d-separation, that $Y_1 \rightleftharpoons \cdots \rightleftharpoons Y_m$ is active in $\mathcal{H}$ given $\mathbf{Z}$ implies that $Y_m \rightleftharpoons \cdots \rightleftharpoons Y_m$ is active in $\mathcal{G}$ given $\mathbf{Z}$, for an arbitrary trail.

If this were true, then we could factor $p$ as,

$$p(\mathbf{y}) = \prod_{i=1}^{n} p(y_i \mid y_1, \ldots, y_{i-1})$$
$$= \prod_{i=1}^{n} p(y_i \mid \{y_1, \ldots, y_{i-1}\} \cap \text{Markov}_{\mathcal{G}}(y_i) \cap \mathbb{I}(y_i)),$$

where, $\mathbb{I}(y_i) = \mathbf{z}$ if $y_i \in \mathbf{z}$ and $\mathbf{y}$ otherwise, is defined to prevent edges from latent nodes into observed ones.

The problem is in going from the first to the second line. For example, consider the factor for an arbitrary latent node, $Z_i$. We have not conditioned on its *complete* Markov blanket—only the children, and parents of children that occur previously in the ordering—and so we cannot assert that $Z_i$ is independent from all the other previous variables.

It is easy to construct counterexamples, for which the influence of a variable flows through one of its parents to effect another variable prior in the ordering that has not been conditioned on. For instance, see Figure 2.

Consider our first example in parts (a-b). The heuristic inverse, $\mathcal{H}$, in (b) asserts that $B \perp C$, since any path between the two variables is blocked by the v-structure. However, $B \perp C$ does not hold in the model, $\mathcal{G}$, in (a), as the path $B \leftarrow A \rightarrow C$ is active. As $\mathcal{H}$ asserts a conditional independence relationship that does not hold in $\mathcal{G}$, it is not faithful to the model. A similar argument can be produced for the second example in parts (d-e). A correct inverse structure produced by our algorithm is given in parts (c) and (f).

# D   Details of experimental setup

Optimization was performed with Adam (Kingma & Ba, 2014) and the default hyperparameters, $\beta_1 = 0.9$ and $\beta_2 = 0.999$.

## D.1   Relaxed Bernoulli VAEs

We perform amortized SVI on a relaxed SBN with 30 latent units on the MNIST data set that has been statically binarized, and use the standard $50,000/10,000/10,000$ split for train/test/validation. The relaxed Bernoulli prior had parameter $p = 0.5$ and temperature $\tau = 1/2$, and the relaxed Bernoulli distribution in the inference program, temperature $\tau = 2/3$

A learning rate of `1e-4` was used, with batch size 100.

In the forward model, $p(\mathbf{x} \mid \mathbf{z})$, the parameters were calculated by a tanh feedforward network with two hidden layers of size $[200, 200]$. For the ten mean-field inference programs, the same form of feedforward network was used, varying the size of the hidden layers from $[100, 100]$, $[200, 200]$,...,$[1000, 1000]$. The ten minimally faithful/fully connected inverses were parametrized similarly, adjusting upwards the size of the different hidden layers to match the number of parameters to the corresponding mean-field program.

The annealed importance sampling (AIS) estimate of $\ln(p(\mathbf{x}))$ averaged 5 chains of 5000 intermediate distributions. As in Maddison et al. (2017, C.3), the latents are treated in the logistic space rather than the relaxed Bernoulli space for numerical stability. We found this was also essential for applying annealed importance sampling.

## D.2   Binary tree Gaussian BNs

We model binary tree Gaussian BNs of depth $d$ with distribution, $X_0 \sim N(0, 1)$, $X_i \mid x_{\lfloor (i-1)/2 \rfloor} = y \sim N(w_i y, 1)$,  $i = 1, \ldots, 2^d - 2$, where the $\{w_i\}$ are fixed constants sampled from $U[1/2, 2]$ and we treat the leaves $\{x_{2^{d-1}-1}, \ldots, x_{2^d-2}\}$ as the observed variables.

In both the heuristic/Stuhlmüller's method and most compact inference programs, each inverse factor was parametrized with a normal distribution using a two hidden-layer ReLU feedforward network with $[100, 100]$ and $[97, 97]$ hidden units, respectively, to map from its parents to the distribution parameters.

A ReLU feedforward network with two hidden layers was also used for the fully connected and natural minimally faithful inference programs, with $[501, 501]$ and $[1210, 1210]$ hidden units, respectively. The MADE masks reduce the effective number of parameters, explaining why these numbers are greater than that for the heuristic inference program.

The total number of parameters for the heuristic, fully connected, most compact, and natural inference programs were 160545, 159136, 156021, and 159901, respectively.

The learning rate was initialized to `1e-3`, decimating when learning converged, for example, every 100 epochs in the case of $d = 5$. A batchsize of 250 was used, new samples from the generative

model being drawn every minibatch for training, with 10 minibatches considered to constitute an epoch, and the test objective evaluated on a single minibatch every epoch.

The exact posterior under the true factorization can be calculated by using the equivalence between Gaussian BNs and multivariate normal distributions (Koller & Friedman, 2009, §7.2)—first the forward model is converted to the parameters of a multivariate normal distribution using Theorem 7.3, which is then transformed back into a Gaussian BN for the posterior using our true factorization and Theorem 7.4. Samples from the posterior can be drawn by ancestral sampling.

We evaluate inference amortization by calculating the average log-posterior of a minibatch from the encoders every epoch under five fixed datasets of the observed variables (which have not be seen by the optimizer).

### D.3 Bayesian Gaussian Mixture Models

We model a Bayesian Gaussian mixture model with $K = 3$ clusters and $N = 200$ two-dimensional samples. The variance parameters of the clusters were parametrized with $\sigma_{1k}, \sigma_{2k}, \rho_k$, where $\rho_k$ is the correlation between the two dimensions. The inference network terms with distributions over vectors were parametrized by MADE, and each inverse factor was parametrized with a suitable probability distribution—$\phi$ with a Dirichlet, $\rho_k$ with Kumaraswamy, $\mu$ with a mixture of Gaussians, $\sigma_{1k}$ and $\sigma_{2k}$ with Inverse Gamma distributions, and $z$ with a Categorical.

The MADEs constituted of two hidden-layer ReLU feedforward network with 360 hidden units per layer for the NaMI inverse and 50 for the fully connected inverse, so that the total number of parameters in the network would be held fixed to allow for a fair comparison. The total number of parameters for the fully connected and natural inference programs were $820047$, and $826779$, respectively.

The learning rate was initialized to `1e-3` and Adam algorithm was used. A dataset of 2000 samples was sampled from the generative model for training the inference network, in minibatches of 200. When the validation error decreased, a new dataset was drawn and training continued.

### D.4 Minimal and Non-minimal Faithful Inverses

The setup for this experiment was as per D.2 unless stated otherwise. We used a model of depth $d = 4$ rather than $d = 5$, parametrizing the forward-NaMI inverse with separate networks for each conditional distribution, rather than a single tree-MADE network. This was because adding extra edges to forward-NaMI broke the ability to share weights, and we wanted the same parametrization scheme for all three inverses. Each network had two hidden layers of size 100. The two inverses with additional edges over the forward-NaMI one used networks with two hidden layers of size 99 in order to keep the total capacity roughly fixed.

## E  Neural density estimators for weight-sharing

### E.1  MADE

We use the masked autoencoder distribution estimator (MADE) model (Germain et al., 2015) extended for conditional distributions (Paige & Wood, 2016) to model fully connected distributions over latent variables, conditioning on all observations, that is,

$$q(\mathbf{z} \mid \mathbf{x}) = \prod_{i=0}^{m-1} q_i(z_i \mid z_1, \ldots, z_{i-1}, \mathbf{x}).$$

From a high level, MADE works by using a single feedforward network that takes as inputs $(\mathbf{x}, \mathbf{z})$, and outputs parameters of all the factors $\{q_i\}$. The weights of the feedforward network are multiplied elementwise by masking matrices so that if one were to trace a path back from an output parameter for $q_i$ to the inputs, that parameter would only be connected to $\{z_1, \ldots, z_{i-1}, \mathbf{x}\}$.

To make things more concrete, consider a single-hidden-layer feedforward network, used to the calculated the parameters, $\theta$, of binary valued data,

$$\mathbf{h} = \sigma_w \left( \mathbf{b} + (W \odot M^{(w)})(\mathbf{z}, \mathbf{x}) \right)$$

$$\theta = \sigma_v \left( \mathbf{c} + (V \odot M^{(v)})\mathbf{h} \right),$$

where $\mathbf{b}, \mathbf{c}, W, V$ are real-valued parameters to be learned, $\odot$ denotes elementwise multiplication, $\sigma_w, \sigma_v$ are nonlinear functions, and $M_w, M_v$ are fixed binary masks.

To each hidden unit, $h_i$, we assign an integer uniformly from $\{1, \ldots, m-1\}$. To each input unit we assign the integer $0$ if it corresponds to an observation, $x_i$, and the integer $i$ if it corresponds to the latent unit $z_j$. The input mask element $M_{i,j}^{(w)}$ represent a connection from the $i$th input unit to the $j$th hidden unit. Thus we set $M_{i,j}^{(w)} = 1$ only when the integer assigned to the $i$th input is less than the integer assigned to the $j$th hidden unit, and $0$ otherwise. In this way, if the $j$th hidden unit is assigned $k$, it will depend on $\{z_1, \ldots, z_{k-1}, \mathbf{x}\}$. The output mask $M^{(v)}$ is constructed similarly by assigning the integer $i$ the units corresponding to the parameters of $q_i$.

This method can be easily extended to feedforward networks with more than one hidden layer. For instance, if there is a second hidden layer $\mathbf{h}'$ with mask $M^{(w')}$, we assign each hidden unit $h_i'$ an integer uniformly from $\{1, \ldots, m-1\}$ (or in fact, we can start from the lowest integer assigned to an $h_i$), and set $M_{i,j}^{(W')} = 1$ only when the integer assigned to $h_i$ is less than or equal to the integer assigned to $h_j'$. In this way, if $h_j'$ is assigned integer $k$, it depends on $\{z_1, \ldots, z_{k-1}, \mathbf{x}\}$ through hidden units $\{h_i\}$ assigned $k$, it depends on $\{z_1, \ldots, z_k - 2\}$ through hidden units $\{h_i\}$ assigned $k-1$, and so on. This is a form of weight sharing.

We use two hidden layer MADEs in our experiments, including, in addition, masked skip-weights from the inputs to the outputs, as is recommended in Germain et al. (2015).

## E.2 Tree MADE

In trying to model the regular but less-than-fully-connected dependency structure of minimally faithful inverses to binary trees, we had the following novel insight. Rather than thinking of the integers assigned to the input, hidden, and output units as simply numbers, we recognize that they actually identify subsets of the model variables. That is, $k$ corresponds to $\{z_0, \ldots, z_{k-1}, \mathbf{x}\}$. The mask weight is set to $1$ only when the first subset is contained in the second. A difference choice of subsets will allow us to model another dependency structure, with the subset inclusion relationship defining weight sharing across the factors.

Running our algorithm on the binary tree Gaussian network of §3.2, reveals that one minimally faithful inverse for a model of depth $d$ comprises factors,

$$q_i(x_i \mid x_{i+1}, \ldots, x_{2(i+1)}), \quad i = 0, 1, \ldots, 2^d - 2.$$

We break up the subsets $\{x_{i+1}, \ldots, x_{2(i+1)}\}$ into,

$$\{x_{i+1}\},$$
$$\{x_{i+2}, \ldots, x_{2(i+1)}\},$$
$$\{x_{i+3}, \ldots, x_{2(i+1)}\},$$
$$\vdots$$
$$\{x_{2i+1}, \ldots, x_{2(i+1)}\}$$

for each $i$, and assign each a unique integer. The hidden units are uniformly assigned one of these subsets. The input unit for $x_i$ is assigned the subset $\{x_i\}$ and the output units for the parameters of $q_i$ are assigned the subset $\{x_{i+1} \ldots, x_{2(i+1)}\}$. The mask from one hidden, input, or output unit to another is set to $1$ only when the subset corresponding to the first unit is contained in the subset corresponding to the second unit.

By construction, this feedforward network will give the parameters for the $\{q_i\}$ such that $q(\mathbf{z} \mid \mathbf{x})$ is a minimal I-map for the posterior. This idea can clearly be generalized to arbitrary dependency

structures, which we leave for future work. We can algorithmically determine the form of the inverse factors in a minimally faithful inverse offline, extract all subsets of their scopes, and perform the same procedure as above.

# F  Theory

Here, we examine the complexity of the inversion problem and prove the correctness of NaMI's graph inversion.

## F.1  Inversion complexity

To understand the theoretical gains we obtain, it is useful to compare it with a simpler, but suboptimal, alternate that uses the d-separation properties of a BN structure to form a minimally faithful inverse. By the general product rule, any distribution over $\mathbf{y} = \{y_1, \ldots, y_n\}$ can be factored as $p(\mathbf{y}) = \prod_{i=1}^{n} p(y_i \mid y_1, \ldots, y_{i-1})$, for any ordering of $\mathbf{y}$. The conditioning sets, $\{y_1, \ldots, y_{i-1}\}$, can be restricted according to the conditional independence assertions made by $p$. To produce a minimal I-map, they can be restricted as $p(\mathbf{y}) = \prod_{i=1}^{n} p(y_i \mid \tilde{\mathbf{y}}_i \subseteq \{y_1, \ldots, y_{i-1}\})$ where $\tilde{\mathbf{y}}_i$ is a minimal subset such that $y_i \perp (\{y_1, \ldots, y_{i-1}\} \setminus \tilde{\mathbf{y}}_i) \mid \tilde{\mathbf{y}}_i$.

Consequently, one could instead produce a minimally faithful inverse for $p(\mathbf{z} \mid \mathbf{x})p(\mathbf{x})$ as follows. Set $\mathbf{y} = (\mathbf{z}, \mathbf{x})$ to have an arbitrary topological ordering on $\mathbf{z}$ and $\mathbf{x}$, separately. Initialize $\tilde{\mathbf{y}}_i = \{y_1, \ldots, y_{i-1}\}$. Scan through $y_j \in \tilde{\mathbf{y}}_i$, removing each one if $y_i \perp y_j \mid \tilde{\mathbf{y}}_i \setminus \{y_j\}$, repeating until none can be removed and $\tilde{\mathbf{y}}_i$ is a minimal subset.

In the worst case for this naive approach, we must scan through $O(n^2)$ variables $n$ times, and the cost of determining whether to remove a variable from $\tilde{\mathbf{y}}_i$ is $O(n)$ (Koller & Friedman, 2009, Algorithm 3.1). Thus, this naive method has running time $O(n^4)$. NaMI's graph reversal, in contrast has a running time of order $O(nc)$ where $n$ is the number of variables in the graph and $c << n$ is the size of the largest clique in the induced graph.

## F.2  Proof of correctness

**Theorem 2.** *The Natural Minimal I-Map Generator of Algorithm 1 produces inverse factorizations that are natural and minimally faithful.*

*Proof.* As in the main paper, let $\tau$ denote the reverse of the order in which variables were selected for elimination such that $\tau$ is a permutation of $1, \ldots, n$ and $\tau(n)$ is the first variable eliminated.

We first show that inverse structure $\mathcal{H}$ produced by Algorithm 1 is guaranteed to be a valid inverse factorization, that is, it factors as

$$q_\psi(\mathbf{z} \mid \mathbf{x}) = \prod_{i=1}^{n} q_i(z_{\tau(i)} \mid \mathrm{Pa}_{\mathcal{H}}(z_{\tau(i)})) = \prod_{i=1}^{n} q_i(z_{\tau(i)} \mid \mathrm{Pa}_{\mathcal{H}}(z_{\tau(i)}), \mathbf{x}) \tag{1}$$

where $\mathbf{z}$ and $\mathbf{x}$ are the observed and latent variables, and $\mathrm{Pa}_{\mathcal{H}}(z_{\tau(i)}) \subseteq \{\mathbf{x}, z_{\tau(1)}, \ldots, z_{\tau(i-1)}\}$ indicates the parents of $z_{\tau(i)}$ in $\mathcal{H}$. There are two critical features this factorization encapsulates that we need to demonstrate to show $\mathcal{H}$ provides a valid inverse factorization: $\mathbf{x}$ only appears in the conditioning variables (i.e. there are no density terms over observations) and all terms can, if desired, be conditioned on the full set of observations.

The former of these straightforwardly always holds, since we only add edges *into* latent variables when the inverse, $\mathcal{H}$, is constructed (Line 11 in Algorithm 1). Therefore, the algorithm can never add in edges *to* an observed node.

The latter is more subtle, as NaMI may produce factors which are not explicitly conditioned on all the observations. However, because, as we demonstrate later, the inversion is faithful, the corresponding $z_{\tau(i)}$ must be conditionally independent of the all observations which are not parent nodes, given the state of the parent nodes. In other words, if the inversion is faithful, this ensures that each $q_i(z_{\tau(i)} \mid \mathrm{Pa}_{\mathcal{H}}(z_{\tau(i)})) = q_i(z_{\tau(i)} \mid \mathrm{Pa}_{\mathcal{H}}(z_{\tau(i)}), \mathbf{x})$, and thus that we have a valid inverse factorization.

Next, we prove that Algorithm 1 produces natural inverses. A general observation is that if $z_i$ is eliminated after $z_j$, there cannot be an edge from $z_j$ to $z_i$ in $\mathcal{H}$. When the algorithm is run

in topological mode, variable elimination is simulated in a topological ordering, and so all of a variable's descendants are eliminated after it is. Therefore there cannot be an edge from a variable to its descendant, and hence the factorization is natural. An equivalent argument applies when the algorithm is run in the reverse topological mode.

Finally, we prove that the inverse factorization is minimally faithful. At a high-level, our proof consists of showing an equivalence to a process where we start with a fully connected graph over the variables and sequentially prune edges in the graph according to the independencies revealed by the clique tree generated from simulating variable elimination, terminating when no more edges can be pruned. By showing that each individual pruning never induces an unfaithful independence, we are able to demonstrate that the graphs at each iteration of this process—including the final inverse graph—is faithful, while minimality follows from the fact that the process terminates when it is not possible to prune any given edge.

More precisely, by the general product rule, $p(\mathbf{z}|\mathbf{x}) = \prod_{i=1}^{n} p(z_{\tau(i)}|z_{\tau(<i)}, \mathbf{x})$, where $z_{\tau(<i)} = \{z_{\tau(1)}, \ldots, z_{\tau(i-1)}\}$, for any possible $\tau$, and any graph with this factorization is an I-map for the posterior. Each term can be simplified according to the conditional independencies encoded by the posterior and the corresponding graph will still be an I-map for the posterior. For instance, if $z_{\tau(i)}$ is independent of $\{z_{\tau(1)}, \ldots, z_{\tau(i-2)}\}$ given $\{z_{\tau(i-1)}, \mathbf{x}\}$, then $p(z_{\tau(i)}|z_{\tau(<i)}, \mathbf{x}) = p(z_{\tau(i)}|z_{\tau(i-1)}, \mathbf{x})$. By definition, the variable elimination is run in the opposite order to $\tau$. This produces a unique corresponding clique tree (see Appendix A.5). Furthermore, because we introduce a new factor at each iteration, the number of cliques in this clique tree matches the number of latent variables in the original BN, with each clique being associated with the corresponding variable that was eliminated at that iteration (though the clique itself may contain multiple variables). We can thus define $C_{\tau(i)}$ as the unique clique corresponding to the elimination of $z_{(\tau(i))}$. Further, we can define $S_{\tau(i)}$ as the sepset between $C_{\tau(i)}$ and $C_{\tau(i+1)}$, i.e. the set of variables shared between these two cliques. By the correspondence between clique trees and induced graphs, $S_{\tau(i)}$ is exactly the unmarked neighbours of $z_{\tau(i)}$ in the partially constructed induced graph at step $n + 1 - i$. Therefore, setting the parents of $z_{\tau(i)}$ to be its unmarked neighbours in Line 11 of Algorithm 1 constructs $\mathcal{H}$ with the factorization

$$q_\psi(\mathbf{z} \mid \mathbf{x}) = \prod_{i=1}^{n} q_i(z_{\tau(i)} \mid S_{\tau(i)}), \tag{2}$$

which is of the form of (1) with $\mathrm{Pa}_{\mathcal{H}}(z_{\tau(i)}) = S_{\tau(i)}$.

By construction, all $z_{\tau(>i)} = \{z_{\tau(i+1)}, \ldots, z_{\tau(n)}\}$ are upstream in the clique tree from $z_{\tau(i)}$ (and thus downstream in the factorization), meaning they will never be included by $S_{\tau(i)}$. Furthermore, the sepset property of clique trees (Koller & Friedman, 2009, Theorem 10.2) guarantees that $z_{\tau(i)}$ is independent from $z_{\tau(<i)} \setminus S_{\tau(i)}$ given $S_{\tau(i)}$. Therefore, we have that $p(z_{\tau(i)} \mid S_{\tau(i)}) = p(z_{\tau(i)} \mid z_{\tau(<i)}, \mathbf{x})$ for each variable and so

$$p(\mathbf{z} \mid \mathbf{x}) = \prod_{i=1}^{n} p(z_{\tau(i)} \mid z_{\tau(<i)}, \mathbf{x}) = \prod_{i=1}^{n} p(z_{\tau(i)} \mid S_{\tau(i)}). \tag{3}$$

This is the same as the factorization produced by NaMI, as given in (2), and so we conclude that $\mathcal{H}$ is an I-Map of $\mathcal{G}$ and thus a faithful inverse.

The minimality now follows from the fact that the sepset $S_{\tau(i)}$ is also the minimal separating set (see, e.g., the proof of (Koller & Friedman, 2009, Theorem 4.12)). In other words, for each $i$, there is no $T_i \subsetneq S_{\tau(i)}$ such that $p(z_{\tau(i)} \mid T_i) = p(z_{\tau(i)} \mid S_{\tau(i)})$. Suppose we were to remove an edge, $z_{\tau(i)} \leftarrow y_j$, from $\mathcal{H}$, where $y_j \in \{\mathbf{x}, z_{\tau(<i)}\}$ (remembering that by construction we have no edges from $z_{\tau(>i)}$ to $z_{\tau(i)}$). This edge would have been constructed due to sepset $S_{\tau(i)}$. If removing the edge did not make $\mathcal{H}$ unfaithful to $\mathcal{G}$, then this would imply that $p(z_{\tau(i)} \mid S_{\tau(i)} \setminus \{y_j\}) = p(z_{\tau(i)} \mid S_{\tau(i)})$ as none of the other factors will change. But we have already shown this is not possible. Hence, by contradiction, $\mathcal{H}$ is minimally faithful. $\square$