[Reviews · NeurIPS 2018]

Reviewer 1



This paper proposed a graph structure based on the generative model for post inference of the latent variables. The proposed structure keeps the dependence and independence among latent variables, and tries to use connections as less as possible. The work is meaningful in inference of graphical model and the results are promising. I tend to accept this paper. Still some concerns are listed: 1. Definition 3 is confusing. 2. In the experiments of section 3.1, What is the difference between the standard VAE inference (simply invert the edges of the graph) and the minimally faithful inverse of the VAE? The proposed method is only compared with the mean-field case, how about comparing with the standard VAE inference case (simply invert the edges of the graph)? 3. What is the dimension of the data used in section 3.2? 4. The proof for the correctness of the method is confusing.

Reviewer 2



Targeting the problem of automatic amortized variational inference, the authors proposed a new graph inversion method named NaMI. Given a generative model, NaMI is proved to be capable of constructing a faithful, minimal, and natural inverse graph. Accordingly, NaMI is directly applicable to automatically provide the structure of the reference model in variational inference. The proposed method could be of great use in complicated practical variational inference applications. The quality and clarity is fine. As far as I know, the proposed method is original. Therefore, I vote for acceptance.

Reviewer 3



The authors present a novel way to automatically generate the dependency structure for constructing inference network. They frame the problem as a graphical-model-inversion problem and propose to use variable elimination and the min-fill heuristic. The proposed method generates a faithful and minimal structure for constructing inference network. The authors show that the generated inference network outperforms the existing heuristically designed network and the fully connected network. This submission is well-written. I enjoying reading this paper. I have the following questions related to this work. (1) About the parametric form of variational factors in continuous cases Since the true posterior distribution is intractable, why do we use the same (distribution) parametric form in the inference network as the one in the model? Does the choice of the parametric form play a role in choosing the dependency structure of inference network? If the parametric form is limited, does a fully connected inference network have more expressive power? (2) For a deep hierarchical Bayesian network, do skip connections play a role in training? Figure 5 shows that skip connections could speed up training at the first epochs. Why is the minimality important? How about a faithful structure with skip connections from observations to latent variables? For example, we can construct a dependency structure by adding these skip connections into a faithful and minimal structure.